# Can Mechanical Strain and Aspect Ratio Be Used to Determine Codominant Unions in Red Maple without Included Bark

Gregory A. Dahle [1,*], Robert T. Eckenrode IV [1], E. Thomas Smiley [2], David DeVallance [3,4] and Ida Holásková [5]

1   Division of Forestry and Natural Resources, Davis College of Agriculture, Natural Resources and Design, West Virginia University, Morgantown, WV 26505, USA; robeckenrode@gmail.com
2   Bartlett Tree Research Laboratories, 13768 Hamilton Rd., Charlotte, NC 28278, USA; tsmiley@bartlett.com
3   InnoRenew CoE, 6310 Izola, Slovenia; david.devallance@upr.si
4   Department of Applied Natural Sciences, University of Primorska, 6000 Koper, Slovenia
5   West Virginia Agriculture and Forestry Experiment Station, Davis College of Agriculture, Natural Resources and Design, West Virginia University, Morgantown, WV 26506, USA; ida.holaskova@mail.wvu.edu
*   Correspondence: gregory.dahle@mail.wvu.edu; Tel.: +1-304-293-6292

**Abstract:** Arborists maintain trees in landscapes where failure can cause damage to infrastructure. Codominant branch unions are considered less stable than lateral branch unions. Previous research has found that unions can be considered codominant when aspect ratio is greater than 0.70 when included bark is present, yet it remains unclear if this threshold is reasonable in the absences of included bark. We utilized digital image correlation to measure strain (deformation) and separation angle to failure to better understand how mechanical loads move through *Acer rubrum* L. (red maple) branch unions. Strain was found to be higher in the branch regions in limb failure and ball and socket failure modes and strain was greater in the branch protection zone regions of imbedded and flat failures. Strain at failure was found to decrease with increasing aspect ratio, plateauing beyond aspect ratios of 0.83. In the absence of included bark, red maple branch unions appear to become codominant at aspect ratio of 0.83. We recommend that arborists should proactively manage to keep aspect ratios lower than 0.60 and consider mitigation options as aspect ratios approach 0.70.

**Keywords:** aspect ratio; arboriculture; biomechanics; digital image correlation; red maple; strain

## 1. Introduction

Arborists and urban forest managers maintain trees in a landscape that is dominated by humans where tree failure (whole tree or portions of a tree) can cause damage to infrastructure, personal property, or personal injury [1–3]. Many of these failures occur during heavy or extreme storm events such hurricanes, ice storms or heavy snow events [1,4,5]. The frequency of extreme weather events is increasing due to climate change [6]. Thus, to reduce failure, urban forest managers need more information on which trees have an increased likelihood of failure [1,7,8].

The strength of codominant unions may sometimes be questionable. Research has found that codominant unions are less stable than lateral branch unions [9–13]. Codominant stems arise from simultaneous vegetative development of axillary buds at the branch apex or from simultaneous development of collateral buds (dormant or adventitious), likely after the loss of apical/terminal end of the main growth axis [13]. Additionally, species that have opposite branch arrangements are prone to codominant branching [14,15]. Being able to identify less stable branch unions more accurately would allow urban forest managers to reduce failures prior to pending extreme storm events.

Some codominant branch unions have been shown to have lower load-bearing capacity than lateral branch unions [9–13], thus arborists often considered codominant unions to have a higher likelihood of failure than lateral branches. When included bark is present, the union is considered less stable [12,16–18] due to inherent structural defect and lack of cross-lamination

of grain between branch and stem wood. An inverse relationship between attachment stability and aspect ratio has been reported in red maple (*Acer rubrum* L.) [3,12]. Aspect ratio, the ratio of branch diameter divided by stem diameter (measured above the union), is considered an effective predictor of failure stress in static loading trials [3,19–21]. While investigating branch unions from a hydraulic segmentation perspective, Eisner et al. [22] suggested that aspect ratios of 0.75 or above indicate codominance. Kane et al. [21] concluded that unions with aspect ratios greater than 0.70 are codominant, yet these trials include unions with and without included bark. They also found that the failure mode was ball and socket when aspect ratios were less than 0.70, and flat surface or imbedded branch failure when aspect ratio was greater than 0.70 [21]. Yet it remains unclear from a mechanical perspective if the 0.7 threshold for codominant holds for branch unions without included bark. This can be important as in the absence of the readily recognizable included bark, it can be difficult to know when a branch union is codominant and is thus more likely to fail than the more stable lateral union.

Eckenrode et al. [23] noted challenges of using stress formulas to calculate breaking stress due to the irregular geometric shape of branch unions at the point of failure. They also concluded that digital image correlation (DIC) can be used to measure mechanical strain during static loading trials. Strain is a direct result of loading [24–26] and is increasingly being used in arboricultural studies [26–33]. While many practitioners suggest that branch attachment angle is a predictor of union stability, the literature has shown that attachment angle is not correlated with strength [10,11,13,21,34,35]. Kane et al. [21] demonstrated that aspect ratio was a better predictor of attachment stability. Additionally, it is possible that a relationship lies between branch union type (lateral vs. codominant) and separation angle (difference between union angle at failure—initial union angle). Miesbauer et al. [36] discussed a method to measure branch angle using a video recording played on a screen to measure the before and after angles, yet this method was somewhat cumbersome and time consuming. Eckenrode et al. [23] presented a method to measure separation angle during static branch and theorized that unions with larger separation angle would require more loading to induce failure. Separation angle coupled with union failure mode model might provide further insight into the ability of branch unions to bear applied loads. As extreme weather events increase in the future, urban forest managers need more knowledge on which branches are more likely to fail during storms. Knowing which branch unions are more prone to failure will allow managers to apply remedies including pruning, removal, and/or providing supplemental support. This paper presents the use of DIC to examine maximum strain patterns in red maple at varying aspect ratios to better understand the mechanical nature of branch union without included bark. Additionally, this research investigated if maximum strain in the branch union can be predicted by a suite of variables (length, diameters, aspect ratio, separation angle) or failure mode, and if a there is a relationship between maximum strain and separation angle.

## 2. Materials and Methods

Branch unions were collected from the West Virginia University Research Forest (39°40′25.2″ N 79°46′27.9″ ) located in Monongalia County, West Virginia, U.S. The specimens were collected from the top of red maple (*Acer rubrum* L.) trees felled as part of a harvest operation. All specimens were collected on the day that the trees were felled and static load testing was conducted within two days. Branch unions with aspect ratio (branch diameter/stem diameter) of 0.5 and higher were targeted. Branch diameter was taken distal to the branch collar and stem diameter was taken distal to the branch bark ridge. Specimens with included bark or structural damage from felling were excluded from the study. The specimens, 32 in total, came from 28 trees.

Branch length was measured from the branch bark ridge to the terminal bud, by following the curvature of the branch, and recorded to the nearest cm using a 50 m tape. The samples were cut to 50.8 cm long above and below the union as well as the branch, and the cut ends were treated with Packard wood sealant (Packard, Tryon, NC, USA) to

minimize moisture loss prior to testing. Research has shown that moisture contents of excised branch union remain above fiber saturation point for between 5–9 days [37]. To prepare the specimens for the static pull test, a coating of white paint (Rustoleum flat finish) followed by a black speckling (Rustoleum flat finish) was applied to the area of interest. The target size for the speckle was 5–10 pixels, as measured during image capturing by the DIC system.

The DIC system (ARAMIS 3D 5M LT DIC version 6.3.0, GOM Braunschweig, Germany) was calibrated using an ISO-9001 certified calibration panel (350 × 280 mm$^2$) by following the protocol described in Beezley et al. [32] and Dahle [26]. The calibration deviation was less than 0.3% and the facet size was 20 × 20 pixels with a 25% facet overlap. The intersection deviation was set at 0.03% for each stage (stereo photographs) and provided an accuracy of at least 0.3 ± 0.1% in the *x* (horizontal) direction. The working distance was set at 130 cm, and during testing, the DIC system collected 5 frames per second.

The unions were placed in a custom fabricated steel bracket (Figure 1) and secured with ratchet straps. A battery powered 8.9 kN winch (Reese, Plymouth MI) was employed to apply a force to the branch. A rope (19 mm diameter) was attached to the branch 25.4 cm from the union, on the opposite end of the rope, and a bowline on a bight was tied to affix a steel carabineer (50 kN) attached to the steel winch cable. Loading was applied at a constant rate until failure occurred when the wood ruptured. After the static load trials, three cross-sections were obtained to determine moisture content; one from the branch, and one each from the stem wood above and below the union. Specimens were dried at 101 °C until a constant mass was reached. Moisture content was calculated as [(*wet mass-oven dry mass*)/*oven dry mass*].

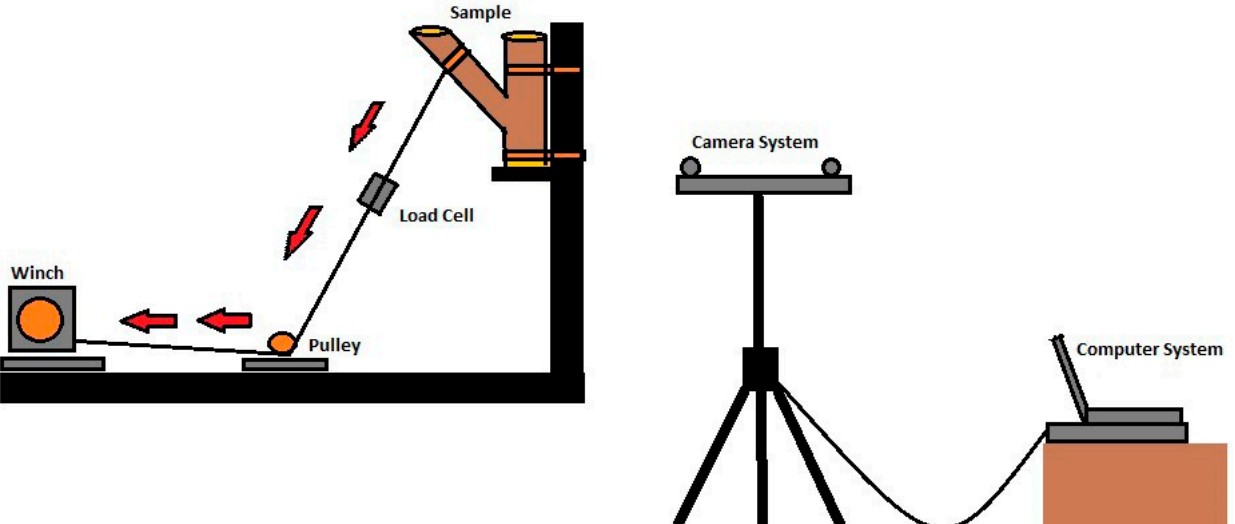

**Figure 1.** A diagram of the static branch pull system. Red arrows indicate direction of winch pull.

The failure stage was obtained by examining DIC stage photos for ultimate failure when the initial tissue rupture was visually observed. Once the data were collected and strain maps computed, sample points were created in the area of interest to connect and analyze strain patterns. Sample points were laid on a grid throughout the area of interest using the grid view feature in the DIC system (Figure 2A). Test points were placed at every fourth facet in the *x* and *y* direction throughout the sampling area. Strain data were obtained for each test point with tensile strain having a positive value and compressive a negative value. All strain data were converted to magnitude by taking the absolute value, and the larger magnitude at failure was labeled as maximum strain.

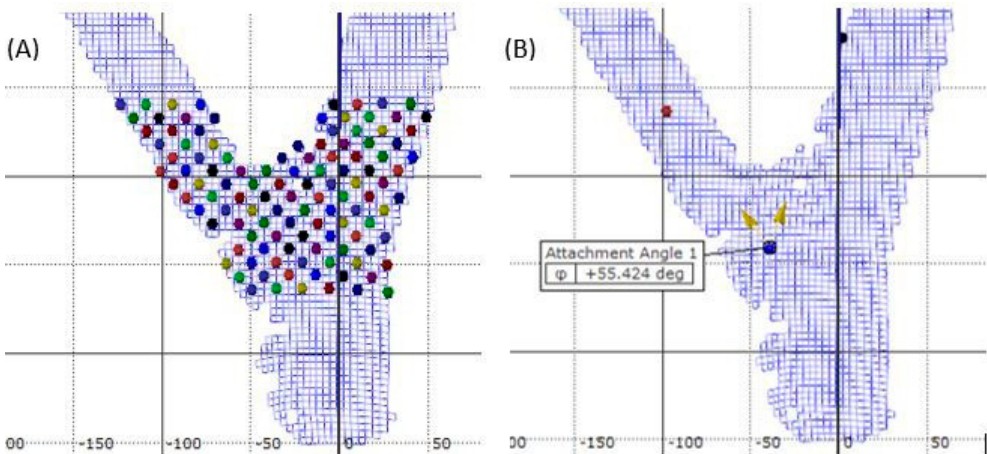

**Figure 2.** Example of the grid system (**A**) of test points used for post-processing measurement of and (**B**) example of the measurement of separation angle using DIC software on codominant red maple.

Test points were grouped into one of seven zones (A–G, Figure 3). To compare unions with each other, the zones created were based on the branch's anatomy. Zone A covered the branch's bottom surface, immediately distal to the attachment. Zone B covered the branch's top half (closer to the stem) with the same dimensions as zone 1. Zone C was defined as the area closest to the union extending upward on the stem to the edge of the area of interest. Zone C extended halfway into the stem (diameter) and ended at the top of the union. Zone D was the area immediately below zone 1 and encompassed the branch's abaxial surface below the union. Zone D was composed of branch tissue in the upper portion and stem tissue in the lower portion and extended over half the branch diameter and down to the bottom of the area of interest. Zone E was immediately proximal to the union from Zone D and extended straight down from the union to the bottom of the area of interest and over to the midpoint of the branch. Zone F was the stem compliment to Zone E and extended half the diameter of the stem. Zone G was below Zone F and extended down to the bottom of the area of interest and halfway across the stem.

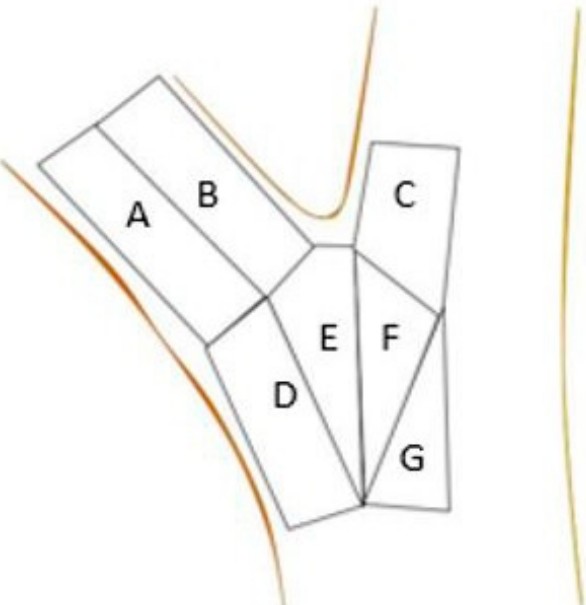

**Figure 3.** Schematic of branch union with the seven zones (A–G) utilized for strain distribution.

Branch attachment angle was measured with the DIC system by placing a reference point in the union where the branch appeared to have originated (Figure 2B). Two vectors

from this reference point were then defined by placing an endpoint on the branch well above the union and an end point on the stem. The subsequent angle between the two vectors was measured before loading (initial angle) and at the point of failure (failure angle). Separation angle was defined as the difference between the failure angle and initial angle [23].

Failures were recorded as one of four types using a system adapted from Kane et al. [21]. The failure types included limb; ball and socket; imbedded; and flat (Figure 4). A limb is classified by the branch tissue failing immediately outside the connection zone leaving no visible stem damage. Ball and socket failures are characterized by excision of stem tissue during failure resulting in a concave shape on the stem's surface. The branch side of the broken surface contains the excised tissue and has a characteristic "ball" shape. An imbedded failure has a slightly convex portion on the branch tissue and a slightly concave portion on the stem tissue, leaving more issue on the stem than a flat failure, while a flat failure is characterized by a cleavage straight down the branch union that leaves nearly equal tissue on the branch and stem. The broken surface's shape is flat and broad and has no concave portion on the stem side.

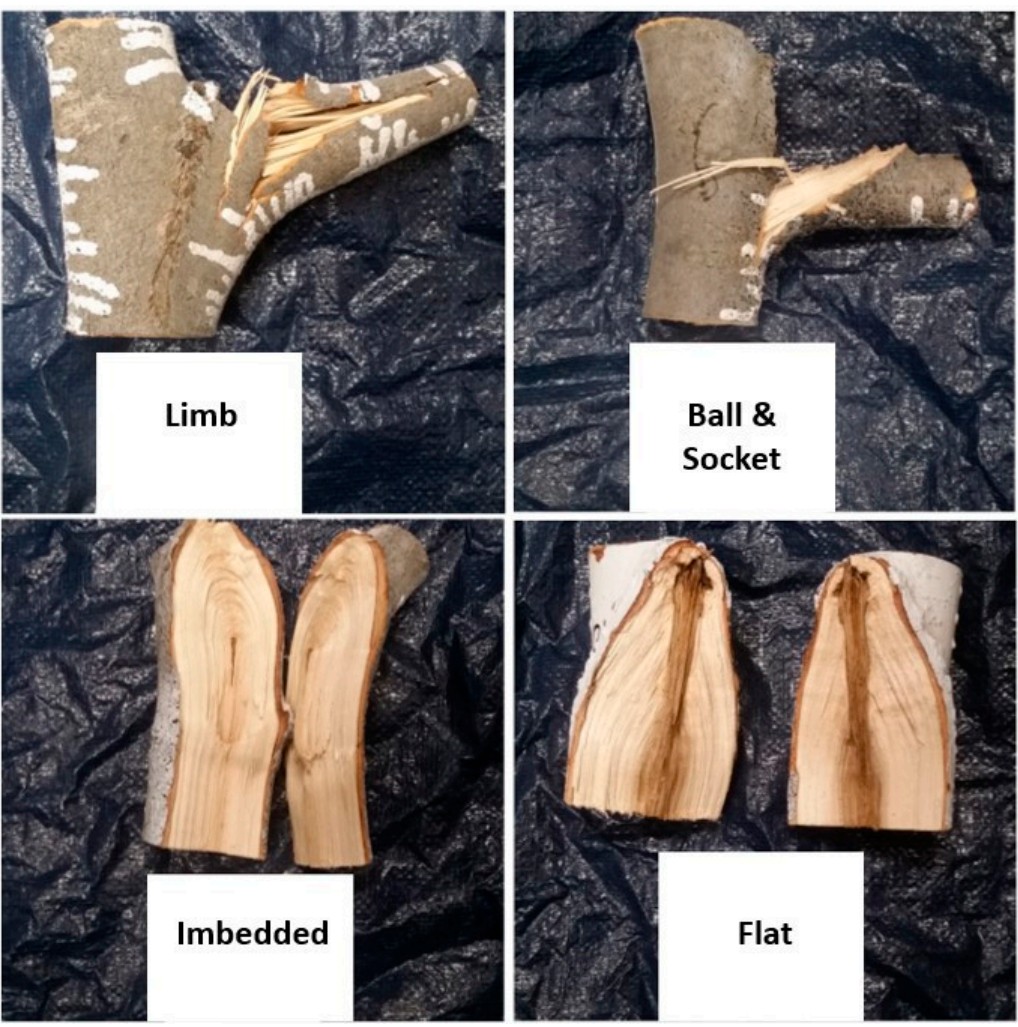

**Figure 4.** Example of failure modes: branch, ball and socket, imbedded, and flat.

To evaluate strain patterns and differences in attachment zones, maximum strain was transformed using $\text{Log}_{10}$ to adjust for normality and labeled $\text{Log}_{10}(\text{MaxStrain})$. The percentage of maximum strain for each failure mode was calculated by dividing the mean strain for each branch zone by the mean maximum strain of the seven zones for the given failure mode. Proc Ttest and Proc GLM with Tukey HSD were used for means

analysis. Exploratory multiple linear regression analysis regressing $Log_{10}$(MaxStrain) on a suite of branch union variables (aspect ratio, separation angle, initial angle, branch length, and branch diameter) was performed with inspection of variance inflation factor (VIF) to prevent multicollinearity. Based on the VIF < 10 and inclusion criteria of each slope parameter having leverage with the maximum *p*-value of 0.1 in the full model, a reduced model contained only aspect ratio and separation angle, with another variation of using the aspect ratio-quantity squared, and the separation angle. In an alternative approach, a generalized regression using the original MaxStrain and a panel of non-normal models, such as $Log_{normal}$, gamma, and Weibull response distributions and BIC criteria, were examined. In addition, the predictive power of aspect ratio and separation angle on maximal strain were also evaluated in individual simple regression. Specifically, the polynomial quadratic fit of $Log_{10}$(MaxStrain) by aspect ratio served for determining the priors (parameter estimators *a*, *b*, and *c*) for the segmented regression with plateau. The proposed functions are

$$Log_{10}(y) = cx_i^2 + bx_i + a \text{ for } x_i \leq x_0, \text{ and}$$

$$Log_{10}(y) = cx_0^2 + bx_0 + a \text{ for } x_i > x_0,$$

which are joined in the knot $x_0$; *a*, *b*, and *c* denote parameter estimators ( intercept, and slopes, respectively), $x_i$ denotes the values of aspect ratio, and *y* is MaxStrain. The knot, $x_0$, or the aspect ratio value for the knot, was unknown and estimated from the data but must satisfy $x_0 = b/2c$ (Kaps and Lamberson, 2004). Proc NLIN of SAS was used to determine the plateau and the knot. For the predictive function of separation angle on maximal strain, a simple linear regression and four-parameter logistic (4PL) were performed. Data were analyzed using JMP and SAS software (JMP®, Version Pro 14.0, SAS Institute Inc., Cary, NC, USA, Copyright ©2015; SAS®, Version 9.4, SAS Institute Inc., Cary, NC, USA, Copyright ©2002–2012). Significance criterion alpha for all tests was 0.05.

## 3. Results

Branch length ranged from 262 to 800 cm with mean of 391 ± 19.6 cm (±SE). Branch diameter ranged from 2.1 to 6.1 cm (*x* = 3.7 ± 0.15 cm), and branch age ranged 6 to 20 years (*x* = 11 ± 0.5 years). Stem diameter above the branch union ranged from 3.0 to 8.6 cm (*x* = 5.0 ± 0.01 cm), and diameter below the union varied from 3.6 cm to 8.7 cm (*x* = 5.9 ± 0.24 cm). Aspect ratio ranged from 0.53 to 0.98 with a mean of 0.76 ± 0.03. Branch diameter differed by failure mode (Table 1), and while the overall model for branch length by failure mode was found to be significant (*p* = 0.0300), the conservative Tukey HSD did not find a difference between the means.

**Table 1.** Summary of branch morphology measurements. Mean (±SE) for branch diameter, length, aspect ratio, and separation angle (failure angle−initial angle). Means with different letters in the column were found to be significantly different using Tukey HSD.

| Failure Mode | Branch Diameter (cm) | Branch Length (cm) | Aspect Ratio | Displacement Angle | N |
|---|---|---|---|---|---|
| Limb | 1.59 ± 0.09 b | 344.00 ± 14.98 a | 0.59 ± 0.02 a | 4.81 ± 0.82 b | 3 |
| Ball and Socket | 1.59 ± 0.12 b | 323.67 ± 11.68 a | 0.64 ± 0.03 a | 2.24 ± 0.42 ab | 9 |
| Imbedded | 1.87 ± 0.07 ab | 389.70 ± 23.95 a | 0.78 ± 0.03 b | 2.08 ± 0.32 a | 10 |
| Flat | 2.20 ± 0.15 a | 465.50 ± 49.02 a | 0.90 ± 0.02 c | 1.57 ± 0.33 a | 10 |
| *p*-Value | 0.0043 | 0.0300 | <0.0001 | 0.0063 | |

Mean branch attachment angle was 46.7° ± 2.3° (range 24.6° to 84.3°), mean failure angle was 49.0° ± 2.3° (range 27.3° to 85.4°), and mean separation angle was 2.3° ± 0.3° (range 0.7° to 6.4°). No statistically significant regression relationship was found between the separation angle and initial angle ($r^2$ = 0.01, *n* = 32), and between the separation angle and failure angle ($r^2$ = 0.00, *n* = 32). No statistically significant relationship was

found between the angle of branch attachment and aspect ratio ($r^2 = 0.0007$, $n = 32$), and a statistically significant relationship was identified between separation angle and aspect ratio ($y = 5.76 - 4.58x$, $r^2 = 0.2159$, $n = 32$); where $y$ = separation angle and $x$ = aspect ratio. A statistically significant relationship was found between separation angle and branch length ($y = 0.36x^2 - 1.9x + 0.65$, $r^2 = 0.3026$, $n = 32$), where $y$ = separation angle and $x$ = branch length. Mean aspect ratio was highest in flat failures with the limb and the ball and socket failure modes having the lowest aspect ratio (Table 1). Separation angle was higher with limb failure and lower in the embedded and flat failures.

Branch moisture content averaged 57% $\pm$ 1.4%. Stem moisture content above the union averaged 62% $\pm$ 1.2%. Mean stem moisture content below the union was 60% $\pm$ 1.8%. Moisture content did not differ between the three locations ($p = 0.1242$, $n = 96$) and was consistently above fiber saturation point of 50%, where material properties of wood are considered constant [26,38–40].

The highest strain for each union was recorded as $Log_{10}$(MaxStrain). Initial multiple least square regression regressed $Log_{10}$(MaxStrain) on aspect ratio, initial angle, separation angle, branch length, and branch diameter. The separation angle and aspect ratio were the strongest candidates for the final model. In an alternative approach, a generalized regression using the original MaxStrain and panel of non-normal models, such as $Log_{normal}$, gamma, and Weibull response distributions and BIC criteria, led to the best fit using mainly the separation angle predictor. Significant simple linear relationship of separation angle and $Log_{10}$(MaxStrain) ($r^2 = 0.39$, slope = 0.14, BIC = 11.7) was superseded by better simple model using 4PL fit ($r^2 = 0.59$, BIC = 5.2). However, significant lower and upper asymptotes in the 4PL model were accompanied by the steep growth rate with high standard error (SE) ($79.99 \pm 11$) as well as high SE associated with inflection point ($3.62 \pm 739.39$). As such, we report the significant second order polynomial fit between $Log_{10}$(MaxStrain) and separation angle and aspect ($y = -0.03x^2 + 0.38x - 0.78$, $R^2 = 0.5658$, $n = 32$, Figure 5), where $y$ = separation angle and $x$ = aspect ratio.

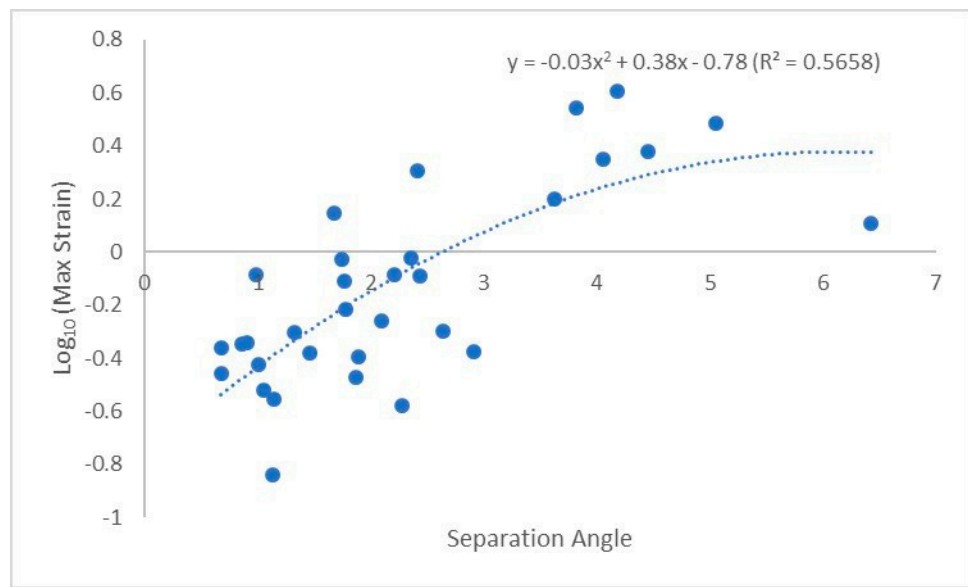

**Figure 5.** Polynomial best fit line between $Log_{10}$(MaxStrain) and separation angle.

No statistically significant relationship was found between the angle of branch attachment $Log_{10}$(Max Strain) ($r^2 = 0.1342$, $n = 32$). $Log_{10}$(MaxStrain) was found to be highest in the limb failure (Table 2, $Log_{10}$(MaxStrain) was back-transformed after analysis). $Log_{10}$ maximum strain for each failure mode was analyzed by zone (Table 3, $Log_{10}$(MaxStrain) was back-transformed after analysis) and generally, the strain was higher in the limb and branch attachment zones (A, B, D, E) with the limb failure and ball and socket failure modes.

**Table 2.** Mean overall max strain $\pm$ SE and (*n*) for each failure mode. Mean with different letters in the column were found to be significantly different using Tukey HSD. Data were analyzed using $\text{Log}_{10}$ but untransformed (original scale) means and SE are used for presentation purposes.

| Failure Mode | Max Strain $\pm$ SE | N |
|---|---|---|
| Limb | $3.46 \pm 1.18$ a | 3 |
| Ball and Socket | $1.38 \pm 0.39$ b | 9 |
| Imbedded | $1.34 \pm 0.21$ b | 10 |
| Flat | $1.08 \pm 0.15$ b | 10 |
| *p*-value | 0.0284 | |

**Table 3.** Mean max strain $\pm$ SE and (*n*) for each branch union zone separated by failure mode. Mean with different letters in the column using Tukey HSD were found to be different. Data were analyzed using $\text{Log}_{10}$ but untransformed (original scale) means and SE are used for presentation purposes.

| Zone | Limb (*n*) | Ball and Socket (*n*) | Imbedded (*n*) | Flat (*n*) |
|---|---|---|---|---|
| A | $3.39 \pm 0.45$ a (16) | $1.24 \pm 0.24$ a (44) | $0.53 \pm 0.06$ bc (56) | $0.59 \pm 0.05$ abc (90) |
| B | $2.36 \pm 0.49$ ab (15) | $0.71 \pm 0.12$ abc (36) | $0.36 \pm 0.03$ c (50) | $0.42 \pm 0.03$ bc (86) |
| C | $0.17 \pm 0.03$ d (14) | $0.11 \pm 0.01$ d (35) | $0.19 \pm 0.01$ d (39) | $0.15 \pm 0.01$ d (56) |
| D | $1.68 \pm 0.25$ ab (13) | $0.9 \pm 0.15$ ab (36) | $0.54 \pm 0.04$ bc (53) | $0.58 \pm 0.06$ abc (66) |
| E | $1.45 \pm 0.2$ b (13) | $0.91 \pm 0.13$ a (30) | $1.14 \pm 0.14$ a (41) | $0.66 \pm 0.06$ ab (72) |

A significant ($R^2 = 0.46$) second-order polynomial was fit between $\text{Log}_{10}(\text{MaxStrain})$ and aspect ratio (Figure 6A), yet at higher aspect ratios $\text{Log}_{10}(\text{MaxStrain})$ appeared to be nearly constant. Spline regression analysis determined that strain plateaued when aspect ratio was greater than 0.83; $\text{Log}_{10}(\text{MaxStrain})$ plateaued at $-0.32$ (Figure 6B). Thus, when aspect ratios were greater than 0.83, the unions were deemed codominant. The regression for $\text{Log}_{10}(\text{MaxStrain})$ below aspect ratio of 0.83 was $y = 8.72x^2 - 14.49x + 5.7$, where $y$ was $\text{Log}_{10}(\text{MaxStrain})$ and $x$ was aspect ratio.

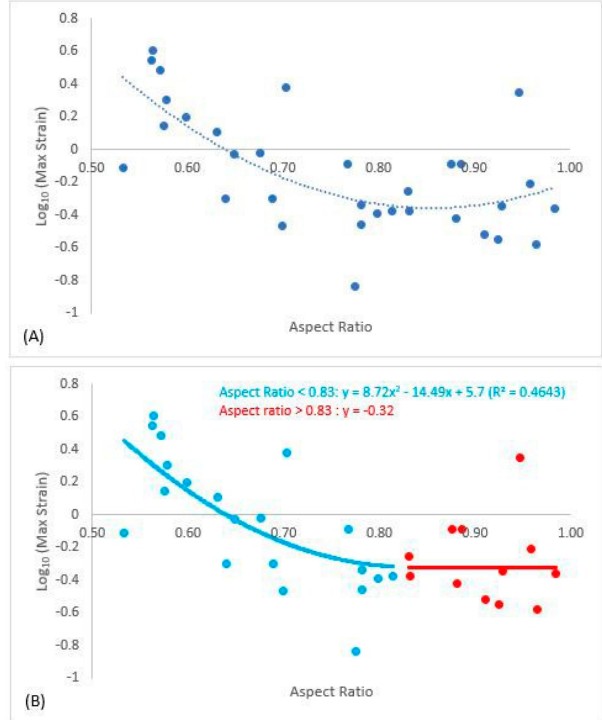

**Figure 6.** Polynomial best fit line between $\text{Log}_{10}(\text{MaxStrain})$ and aspect ratio (**A**). Spline regression determined that the data plateaued when aspect ratio was greater than 0.83. The second graph (**B**) reports the regression equation for $\text{Log}_{10}(\text{MaxStrain})$ and aspect ratio for lateral unions and the plateau value ($-0.32$) for $\text{Log}_{10}(\text{MaxStrain})$ for codominant unions when aspect ratio was greater than 0.83.

## 4. Discussion

The red maple branch union in this study ranged in aspect ratio from 0.5 to 1.0, providing a representative sample of branch unions ranging from lateral to codominant unions. We found that strain decreased with aspect ratio up to 0.83 and then was constant beyond (Figure 6b). The general patterns of strain agreed with previous studies that investigate branch union stability [13,19–21]. Kane et al. [21] reported that branches were codominant after aspect ratio of 0.70 for three species including red maple, yet some of the branch unions in their study had included bark. Given these observations, in the absence of included bark, branch unions appear to have reasonable stability until aspect ratios of 0.83. From a management standpoint in the face of increasing storm events due to climate change, managers should consider branch unions with included bark with aspect ratios greater than 0.83, or even 0.80, as codominant, and between 0.70–0.80 would likely be considered the critical zone in terms of when a union is moving toward codominant. Unions in this critical zone should be treated in order to prevent the loss of stability.

The aspect ratios for the limb (0.59) and the ball and socket (0.65) failure modes were found to be smaller than imbedded (0.78) and flat failures (0.90) (Table 1). This pattern is similar to that reported by Kane et al. [21]. Analysis of strain distribution in our unions lends further insight into how load is transferred through the unions in these static loading trials. In limb and ball and socket failures, mean maximum strain was highest in the branch zones A, B, D, and E (Table 3). In further analysis, the percent of maximum strain was the highest in the underside of the branch (zone A) in terms of peak (100%) mean maximum strain (Figure 7). Loading appeared to move further into the branch union in the imbedded failures and flat failures, as zone E and F encountered higher strains and zone F had peak mean maximum strain. The highest strains in imbedded failures were in the middle of the union (zones E and F), while flat failures strain was more consistent in both the branch attachment zones (A, B, D, E) as well as in zone F of the stem. The pattern of loading might be the result of how woody tissue is laid down during development. Eisner et al. [22] found that branch unions were codominant after aspect ratio of 0.75 in terms of hydraulic segmentation, most likely to the lack of branch and stem tissue overlap. These morphological features are usually absent in codominant stems [14]. The high aspect ratio branches lacked this collar and therefore the ability to isolate strain in response to static loading.

It has been suggested that while ultimate failures during bending of small (1.0 to 2.0 cm) green branches might appear to take place as tensile failures on the upper side of branches, initial failure takes place on the underside in compression [41,42]. This theory is consistent with the knowledge that wood is weaker in compression [43], yet we did not observe any indication of initial compression failure in our study, and the ultimate failure was along the top of the branch or branch union. The specimens in our study were larger than the previous work, and consisted of branch unions with more completed shapes; it is reasonable to see a difference in failure location. Furthermore, it is possible that the overlapping of tissue in the branch collar can lead to shear between the tissues; our tests were not designed to measure shear so we could not determine if shear failures were occurring between branch and stem tissues. A strong positive relationship was found between strain and separation angle (Figure 5). A larger separation angle suggests that the union can absorb more load before failing. This is further evident when looking at failure mode. The limb failure mode had both low aspect ratio and larger separation angle (4.81°).

The separation angle for both the imbedded (2.08°) and flat failures (1.57°) were lower than the limb failures, further suggesting that these unions are less stable. The ball and socket failures (aspect ratio 0.64) had an intermediate separation angle (2.24°) and did not differ from the other three failure modes. Eckenrode et al. [23] found separation angle to be the same in unions with aspect ratio ranging from 0.30 to 0.60. The current study suggests that branch stability might be beginning to shift toward codominance when aspect ratios are beyond 0.65, further suggesting that managers should try to keep aspect ratios lower than 0.60 to ensure a stable branch union.

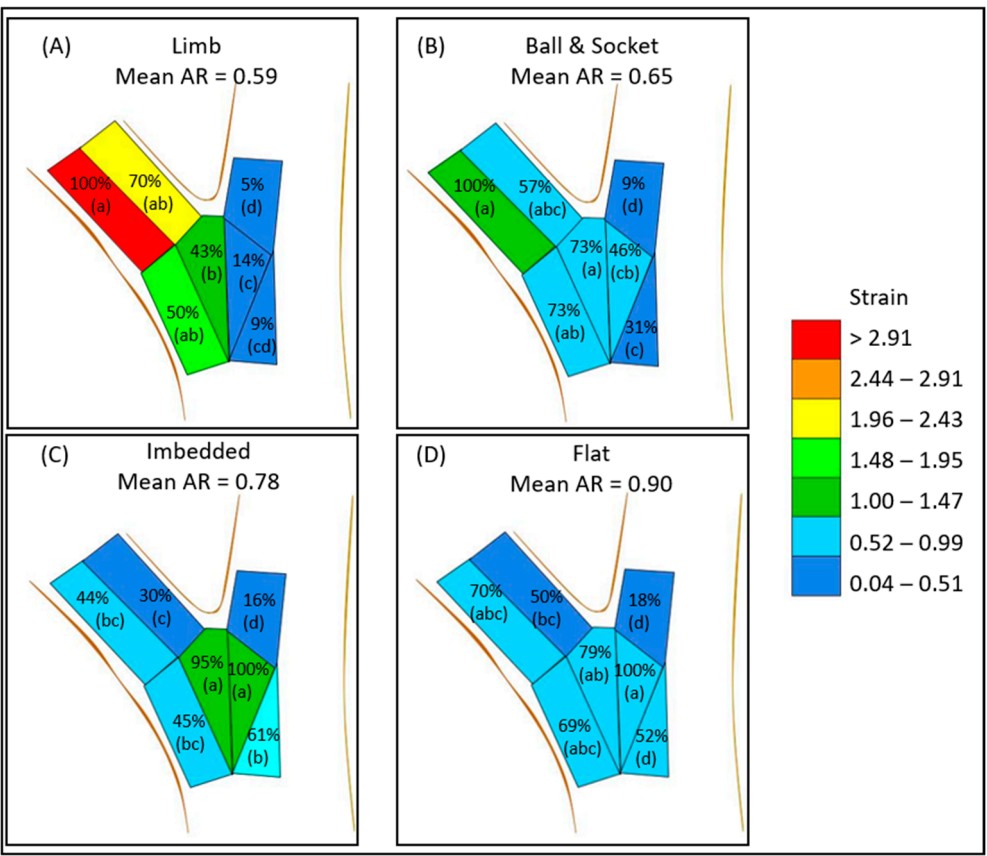

**Figure 7.** Mean max strain distribution and mean aspect ratio (AR) for each failure modes: limb (**A**), ball and socket (**B**), imbedded (**C**), and flat (**D**). Percentages indicate the percent of the mean maximum strain of all the zones for the given failure mode. ANOVA with HSD was conducted within each failure mode, and zones with different letters were found to be significantly different (see Table 3 for numerical analysis).

There was no significant relationship between attachment angle and strain in this experiment. This finding agrees with numerous past studies examining attachment angle and attachment stability [10,11,13,19,34,44,45]. Separation angle not only differs by failure mode; a negative relationship was found with both branch length and branch diameter. Reasons why separation angle was smaller as branches were longer and larger in diameter can be better explained by looking more closely at what happens as branches get longer and increase in diameter. As a branch becomes longer and thicker (diameter), it becomes less flexible [46,47]. This makes intuitive sense and can be seen in nature regularly. Branches become less flexible as the cross-sectional area and moment of inertia increases [24], while at the same time the modulus of rupture and modulus of elasticity of the new wood increases as it transitions from juvenile to mature wood [48–50]. Branches become more rigid as they grow, and their primary function shifts away from sun branches with photosynthetic tissue to structural branches that hold smaller sun branches [47,51–54]. As branch diameter increases, strain induced during bending should also increase, as strain increases with the distance from the pith [5,26,55,56]. We did not test the material properties of the wood in this study; as such, we do not know of the wood in the branch unions' proportion of juvenile to mature wood. While the differences in material properties could be a potential limitation in this study, since branch diameter did not vary greatly in this study, we do not feel that branch size impacted the results. As separation angle was found to vary with branch length and diameter, we feel this is a worthy of additional research as either an explanatory or predicter variable in branch stability.

While we investigated the magnitude of strain, we did not analyze directional strain in this research. Future researchers may wish to consider directional strain to gain additional knowledge on the mechanical nature of the branch union under load. We do not know how these results will translate into larger unions or other species. Another potential limitation is that the branch unions were obtained from trees that were felled during forest operations. While we excluded unions with obvious structural damage, it is possible that some of the branch unions may have suffered internal damage during felling. Finally, this study utilized a static test to induce failure, and many failures in the field take place during dynamic loading events. As such, the results for this study might differ from a dynamic loading trial.

## 5. Implications

This work found that red maple branch union becomes codominant after aspect ratio reaches 0.83. In light of the increased frequency of extreme weather events due to climate change, we recommend using a slightly more conservative threshold of aspect ratio of 0.8 and higher when defining codominant unions in cases where included bark is not present. A conservative threshold may be useful for managers that are concerned with identifying union with higher likelihoods of failure. In terms of young tree development, arborists should proactively manage to keep aspect ratios lower than 0.60 and consider mitigation options as aspect ratios approach 0.70. As aspect ratios move beyond 0.80, union stability appears to be questionable in red maples. This research also found that separation angle may be a useful indicator of attachment stability; however, more research is needed to understand separation angle. This research can aid climbers in choosing a tie-in point, as well as guide pruning decision-making for optimal tree performance and stability. Codominant unions over important targets should be mitigated by removal, supplemental support, or reduction pruning.

**Author Contributions:** G.A.D., R.T.E.IV, E.T.S., D.D. and I.H. contributed to the preparation of this manuscript. Material preparation and data collection were performed by R.T.E.IV, and analysis was conducted by R.T.E.IV, G.A.D. and I.H. The first draft of the manuscript was written by R.T.E.IV and G.A.D., and all authors commented on previous versions of the manuscript. All authors have read and agreed to the published version of the manuscript.

**Funding:** Financial support was received in part from the following sources: National Institute of Food and Agriculture(NIFA) McIntire-Stennis Grant (WVA00108), West Virginia Agriculture and Forestry Experiment Station, West Virginia University Biology Teaching Assistant program. One author, David DeVallance, acknowledges the European Commission for funding the InnoRenew CoE project (Grant Agreement #739574) under the Horizon 2020 Widespread-Teaming program and the Republic of Slovenia (Investment funding of the Republic of Slovenia and the European Union of the European Regional Development Fund) that allowed for efforts related to publication preparation.

**Data Availability Statement:** Not applicable.

**Acknowledgments:** We would like to thank Mark Hoenigman of Busy Bee Service who designed and manufactured the bracket system and John Goodfellow who donated it to the research group. Agatha Dahle created the branch union diagrams and Matt Walker assisted in data analysis.

**Conflicts of Interest:** The authors have no relevant financial or non-financial interests to disclose.

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
