# Peer review of "Can Mechanical Strain and Aspect Ratio Be Used to Determine Codominant Unions in Red Maple without Included Bark"

_forests, doi:10.3390/f13071007_

Round 1
Reviewer 1 Report
The manuscript entitled Determining Codominant Unions in Red Maple Without Included Bark [ID: forests-1766601] describe the study of strain and separation angle of codominant branch union of Red Maple with digital images. The manuscript is well written and needs minor revisions.
- L114: Please provide the information about DIC software. Does the DIC system and DIC software have the same meaning?
- Table 3. : Please begin table with a brief title sentence for the whole table and continue with a short description of what is shown. ‘P-value’ can be move to ‘zone’ column.
- Figure 6. : Duplicated words
- Figure 7. : Duplicated words. / You should define the abbreviation in figure. (ex) AR. / You should explain the meaning of ‘strain’ and ‘number’ of index in legends. Does ‘strain’ mean the Mean data in Table 3 ? Are the data (colors) on each panel related to Table 3?
- Please begin your figure and table legends with a brief title sentence for the whole figure and table, and continue with a short description of what is shown.
Author Response
Reviewer 1
The manuscript entitled Determining Codominant Unions in Red Maple Without Included Bark [ID: forests-1766601] describe the study of strain and separation angle of codominant branch union of Red Maple with digital images. The manuscript is well written and needs minor revisions.
Response: Thank you for the review. We have edited the manuscript to reflect your recommendations.
- L114: Please provide the information about DIC software. Does the DIC system and DIC software have the same meaning?
Response: The system includes the camera, computer and software. For simplicity, we have replaced all “software” with “system” throughout.
- Table 3. : Please begin table with a brief title sentence for the whole table and continue with a short description of what is shown. ‘P-value’ can be move to ‘zone’ column.
Response: Updated
- Figure 6. : Duplicated words
Response: Deleted second “Figure 6”.
- Figure 7. : Duplicated words. / You should define the abbreviation in figure. (ex) AR. / You should explain the meaning of ‘strain’ and ‘number’ of index in legends. Does ‘strain’ mean the Mean data in Table 3 ? Are the data (colors) on each panel related to Table 3?
Response: Duplicated word deleted. Thank you, we have defined AR, and have added more text definition. Actual stain values are found in table 3.
- Please begin your figure and table legends with a brief title sentence for the whole figure and table, and continue with a short description of what is shown.
Response: added

Reviewer 2 Report
General comments:
The study and the written article are valuable contributions to determining the critical aspects ratios in the absence of included bark. This would be useful for practical arborists to determine when to prescribe mitigation options. However, it seems that the immediate title is not very clear. The research work seems to focus on “strain patterns”, “critical aspect ratio” or “strength of codominant unions without included bark”, which imply branch stability, but the title and abstract did not make it clear in the first place and the most important words are listed as keywords (but not the immediate title). I would recommend to add a few words in between “determining” and “codominant”, such as “Determining the ‘Strengths’ in Codominant Unions……
Specific comments:
Title:
It needs to be amended to make it clearer as explained above.
Abstract
Line 1 “codominant branch unions are weaker than lateral ranch unions”. This is an over-simplified and over-generalized statement which may not be useful to explain different scenarios (it may be something like “codominant branch unions may not always be stable…….”). Also, this is not a research question or problem statement. It would be much better to rewrite the first few lines.
It would also be better to mention the “stability” or “strength” of the branch union rather than repeatedly emphasizing “weak” or “failure”. Trees do not inherently perform to be “weak” and tend to “fail”. This would cause misunderstanding and lead to unnecessary over-pruning and tree felling. You are trying to understand the strength of codominant branch union, which may be sometimes questionable (definitely not always questionable).
At the beginning of the abstract, please explain this research is a contribution to the understanding of biomechanical properties of widely planted urban trees and would be a useful guiding reference for arborists.
Keyword:
The authors may consider to add “arborist” /“arboriculture”/ “urban tree management”
Introduction:
Line 20: correct the grammatical errors
Line 21: it would be better to write a few sentences before your Line 21-23. It would be something “ the strength of codominant branch union may be sometimes questionable”. The Line 21-23 should also be re-written to increase phrase clarity.
Line 30: again, the beginning sentence does not seem to prepare the scene for the readers and this may create some wrong impression and confusion. For example, codominant branch unions may not be as strong as people would expect.
Line 31-32: the statement “codominant unions” …“are often” …. “higher likelihood of failure” is just wrong, over-generalized. Please do not make such generalized statement without referring to a specific scenario or situation. are you referring to "with included bark" or "without included bark"? Obviously, not all codominant branch unions without included bark are weak. is it sometimes weak or often weak? (may be 5-10% only)
I did not check all lines regarding the tone and expression. When i read this article, i found it a bit strange to find over-generalized statement and overly "negative" terms and expression. Tree structures are not always problematic. This may be the case for below 5% of all inspected trees. Do not exaggerate the fact.
Results and Discussion:
The research itself should be alright.
Line 253. “Embedded” should be “imbedded”?
Line 314. Grammatical mistakes
Author Response
Reviewer 2
General comments:
The study and the written article are valuable contributions to determining the critical aspects ratios in the absence of included bark. This would be useful for practical arborists to determine when to prescribe mitigation options. However, it seems that the immediate title is not very clear. The research work seems to focus on “strain patterns”, “critical aspect ratio” or “strength of codominant unions without included bark”, which imply branch stability, but the title and abstract did not make it clear in the first place and the most important words are listed as keywords (but not the immediate title). I would recommend to add a few words in between “determining” and “codominant”, such as “Determining the ‘Strengths’ in Codominant Unions……
Response: Thank you for your recommendations, they have improved this manuscript.
Specific comments:
Title:
It needs to be amended to make it clearer as explained above.
Response: We have edited the title to read “CAN MECHANICAL STRAIN AND ASPECT RATION BE USED TO DETERMINE CODOMINANT UNIONS IN RED MAPLE WITHOUT INCLUDED BARK:
Abstract
Line 1 “codominant branch unions are weaker than lateral ranch unions”. This is an over-simplified and over-generalized statement which may not be useful to explain different scenarios (it may be something like “codominant branch unions may not always be stable…….”). Also, this is not a research question or problem statement. It would be much better to rewrite the first few lines.
Response: We changed “weaker” to read as “less stable”
It would also be better to mention the “stability” or “strength” of the branch union rather than repeatedly emphasizing “weak” or “failure”. Trees do not inherently perform to be “weak” and tend to “fail”. This would cause misunderstanding and lead to unnecessary over-pruning and tree felling. You are trying to understand the strength of codominant branch union, which may be sometimes questionable (definitely not always questionable).
Response: Thank you and we agree that stability is a better term than strength. We have changed most of the occurrences of “strength” to read as “stability”, except when referring to specific journal article that used the term “strength”.
At the beginning of the abstract, please explain this research is a contribution to the understanding of biomechanical properties of widely planted urban trees and would be a useful guiding reference for arborists.
Response: A new introductory sentence was added.
Keyword:
The authors may consider to add “arborist” /“arboriculture”/ “urban tree management”
Response: “arboriculture” has been added.
Introduction:
Line 20: correct the grammatical errors
Response: Thank you, we have corrected the errors.
Line 21: it would be better to write a few sentences before your Line 21-23. It would be something “ the strength of codominant branch union may be sometimes questionable”. The Line 21-23 should also be re-written to increase phrase clarity.
Response: Thank you we have added a new initial sentence
Line 30: again, the beginning sentence does not seem to prepare the scene for the readers and this may create some wrong impression and confusion. For example, codominant branch unions may not be as strong as people would expect.
Response: We edited the first and second sentence to soften the statement around codominant unions.
Line 31-32: the statement “codominant unions” …“are often” …. “higher likelihood of failure” is just wrong, over-generalized. Please do not make such generalized statement without referring to a specific scenario or situation. are you referring to "with included bark" or "without included bark"? Obviously, not all codominant branch unions without included bark are weak. is it sometimes weak or often weak? (may be 5-10% only)
Response: We appreciate your comments as you are correct. We edited the first and second sentence to soften the statement around codominant unions.
I did not check all lines regarding the tone and expression. When i read this article, i found it a bit strange to find over-generalized statement and overly "negative" terms and expression. Tree structures are not always problematic. This may be the case for below 5% of all inspected trees. Do not exaggerate the fact.
Response: We have toned down the opening lines.
Results and Discussion:
The research itself should be alright.
Response: Thank you
Line 253. “Embedded” should be “imbedded”?
Response: Thank you for catching this
Line 314. Grammatical mistakes
Response: We added “yet” before “it is possible”.

Reviewer 3 Report
Line 55: From a practical point of view, the angle of ingress of the branch into the tree trunk should not be underestimated.
Line 241: A static branch load test was used in this study. In natural conditions, during strong winds, many force vectors of dynamic nature (including torsion forces) act on trees and their parts. The obtained results are very interesting, but transferring them directly to a practical application will be subject to significant error, which should be borne in mind by the authors.
Line 296: With age in trees, the branches in the pit stop performing their original functions, and there is a gradual death, aging and change in the properties of the wood tissue. This is a natural physiological process, so it is important to describe the height of the location on the tree of the branches studied and their age in order to determine the function they performed. This will allow for a better understanding of the experiment both biomechanically and physiologically.
Since different connections of branches to the trunk were characterized by different strengths/critical angles could the authors determine the biometric characteristics of the trunk and branches, which are good predictors to evaluate the type of damage and thus the strength of the branches.
In my opinion, in the described experiment, a number of features should be additionally provided to better understand the identified and described relationships. Trees are characterized by the so-called adaptive growth, and within one species, they produce differentiated woody tissue. Therefore, the methodological description should be supplemented with:
- the age of the studied trees and the approximate age of the branches/cones used for the experiment,
- the height of the trees used for the study and the height of the branches on the tree used for the study.
- exposure to the wind of the trees used for the study.
In addition, the test trees, depending on their growth conditions, wind exposure, etc., may have developed, for example, tensile wood (which is also found in branches), which significantly affects the strength of the wood tissue. We do not have information on whether the wood tested was verified to be defective. This is very important for the tensile strength of the wood, which we are dealing with in the experiment studied.
The paper is very interesting, and after completing the descriptions of the research, objects will be able to have a wider scientific application and reference in practice.
Author Response
Reviewer 3
Line 55: From a practical point of view, the angle of ingress of the branch into the tree trunk should not be underestimated.
Response: While I inherently feel that branch attachment angle should be important, the literature has yet to find a correlation between branch angle and stability. Both our research team and other researched that we communicate with keep looking to see if a correlation exists.
Line 241: A static branch load test was used in this study. In natural conditions, during strong winds, many force vectors of dynamic nature (including torsion forces) act on trees and their parts. The obtained results are very interesting, but transferring them directly to a practical application will be subject to significant error, which should be borne in mind by the authors.
Response: Agreed. We added two sentences at the end of the discussion section, that indicate a dynamic loading might see different results.
Line 296: With age in trees, the branches in the pit stop performing their original functions, and there is a gradual death, aging and change in the properties of the wood tissue. This is a natural physiological process, so it is important to describe the height of the location on the tree of the branches studied and their age in order to determine the function they performed. This will allow for a better understanding of the experiment both biomechanically and physiologically.
Response: The branches in this study were obtained after a forest harvest, as such we do not have the height of branches as top of the trees were already separated from the trunks.
Since different connections of branches to the trunk were characterized by different strengths/critical angles could the authors determine the biometric characteristics of the trunk and branches, which are good predictors to evaluate the type of damage and thus the strength of the branches.
Response: The aspect ratio was the best predictor of union stability.
In my opinion, in the described experiment, a number of features should be additionally provided to better understand the identified and described relationships. Trees are characterized by the so-called adaptive growth, and within one species, they produce differentiated woody tissue. Therefore, the methodological description should be supplemented with:
- the age of the studied trees and the approximate age of the branches/cones used for the experiment,
Response: As the trunks were not available, we do not have the tree age. We added “branch age ranged 6 to 20 years (x = 11 ± 0.5 years)”, see link 81
- the height of the trees used for the study and the height of the branches on the tree used for the study.
Response: We do not have the overall tree height.
- exposure to the wind of the trees used for the study.
Response: We do not have the wind exposure of the trees.
In addition, the test trees, depending on their growth conditions, wind exposure, etc., may have developed, for example, tensile wood (which is also found in branches), which significantly affects the strength of the wood tissue. We do not have information on whether the wood tested was verified to be defective. This is very important for the tensile strength of the wood, which we are dealing with in the experiment studied.
Response: We did not test the wood for material properties.
The paper is very interesting, and after completing the descriptions of the research, objects will be able to have a wider scientific application and reference in practice.
Response: Thank you.

Reviewer 4 Report
The work presents a study about the use of digital image correlation (DIC) to examine maximum strain patterns in red maple at varying aspect ratios. The research investigates if maximum strain in the branch union can be predicted by certain variables or failure mode, and if a there is a relationship between maximum strain and separation angle. The topic is relevant, and the study was well organized and presented. There are only minor comments to be considered:
In section 2 – Materials and Methods. The description presented in the first paragraphs are too summarized and should be expanded in order to better clarify the applied methodology for the reader. I believe one picture for each paragraph would be helpful. The quality of figure 2 could be improved, specifically considering the resolution.
In section 3 – Results. Tables with all branch length, branch diameter and others investigated parameters would be necessary.
Author Response
Reviewer 4
The work presents a study about the use of digital image correlation (DIC) to examine maximum strain patterns in red maple at varying aspect ratios. The research investigates if maximum strain in the branch union can be predicted by certain variables or failure mode, and if a there is a relationship between maximum strain and separation angle. The topic is relevant, and the study was well organized and presented. There are only minor comments to be considered:
In section 2 – Materials and Methods. The description presented in the first paragraphs are too summarized and should be expanded in order to better clarify the applied methodology for the reader. I believe one picture for each paragraph would be helpful. The quality of figure 2 could be improved, specifically considering the resolution.
Response: Unfortunately, we cannot obtain a higher quality image to replace figure 2. Thank you for the recommendation that we expand the description of the methods section, The description of the DIC process has been described by other researchers including this team in other published articles.
In section 3 – Results. Tables with all branch length, branch diameter and others investigated parameters would be necessary.
Response: We included summary statistic for as well as the range branch length and diameter, as is customary in journal articles.
